# Tuning of Titanium Microfiber Scaffold with UV-Photofunctionalization for Enhanced Osteoblast Affinity and Function

**DOI:** 10.3390/ijms21030738

**Published:** 2020-01-23

**Authors:** Chika Iwasaki, Makoto Hirota, Miyuki Tanaka, Hiroaki Kitajima, Masako Tabuchi, Manabu Ishijima, Wonhee Park, Yoshihiko Sugita, Ken Miyazawa, Shigemi Goto, Takayuki Ikeda, Takahiro Ogawa

**Affiliations:** 1Weintraub Center for Reconstructive Biotechnology, Division of Advanced Prosthodontics, UCLA School of Dentistry, Los Angeles, CA 90095-1668, USA; chikaiwsk@gmail.com (C.I.); hiroaki_k_0315@yahoo.co.jp (H.K.); manabu612@gmail.com (M.I.); drwon69@gmail.com (W.P.); yosshii@dpc.agu.ac.jp (Y.S.);; 2Department of Orthodontics, School of Dentistry, Aichi-Gakuin University, 1-100 Kusumoto-cho, Chikusa-ku, Nagoya, Aichi 464-8650, Japan; machako@dpc.agu.ac.jp (M.T.); miyaken@dpc.agu.ac.jp (K.M.); shig@dpc.aichi-gakuin.ac.jp (S.G.); 3Department of Oral and Maxillofacial Surgery/Orthodontics, Yokohama City University Medical Center, 4-57 Urafune-cho, Minami-ku, Yokohama, Kanagawa 232-0024, Japan; 4Department of Complete Denture Prosthodontics, School of Dentistry, Nihon University, 1-8-13 Kanda-Surugadai, Chiyoda-ku, Tokyo 101-8310, Japan

**Keywords:** acid-etching, ultraviolet treatment, osteoblast, titanium fiber scaffold, titanium surface, cell attachment, osseointegration

## Abstract

Titanium (Ti) is an osteoconductive material that is routinely used as a bulk implant to fix and restore bones and teeth. This study explored the effective use of Ti as a bone engineering scaffold. Challenges to overcome were: (1) difficult liquid/cell infiltration into Ti microfiber scaffolds due to the hydrophobic nature of Ti; and (2) difficult cell attachment on thin and curved Ti microfibers. A recent discovery of UV-photofunctionalization of Ti prompted us to examine its effect on Ti microfiber scaffolds. Scaffolds in disk form were made by weaving grade 4 pure Ti microfibers (125 µm diameter) and half of them were acid-etched to roughen the surface. Some of the scaffolds with original or acid-etched surfaces were further treated by UV light before cell culture. Ti microfiber scaffolds, regardless of the surface type, were hydrophobic and did not allow glycerol/water liquid to infiltrate, whereas, after UV treatment, the scaffolds became hydrophilic and immediately absorbed the liquid. Osteogenic cells from two different origins, derived from the femoral and mandibular bone marrow of rats, were cultured on the scaffolds. The number of cells attached to scaffolds during the early stage of culture within 24 h was 3–10 times greater when the scaffolds were treated with UV. The development of cytoplasmic projections and cytoskeletal, as well as the expression of focal adhesion protein, were exclusively observed on UV-treated scaffolds. Osteoblastic functional phenotypes, such as alkaline phosphatase activity and calcium mineralization, were 2–15 times greater on UV-treated scaffolds, with more pronounced enhancement on acid-etched scaffolds compared to that on the original scaffolds. These effects of UV treatment were associated with a significant reduction in atomic carbon on the Ti microfiber surfaces. In conclusion, UV treatment of Ti microfiber scaffolds tunes their physicochemical properties and effectively enhances the attachment and function of osteoblasts, proposing a new strategy for bone engineering.

## 1. Introduction

Osseointegration of titanium (Ti) materials has been widely utilized in the dental and orthopedic fields. Ti materials are routinely used as a bulk implant to fix and restore bones and teeth. A large and complex material increases the risk of unfavorable events such as foreign body reaction, infection, and mechanical failure, which may interfere with treatment. Bone replacement or bone engineering scaffold is expected to provide a hard tissue rather similar to living tissue to ensure long-term stability. However, a promising material to reconstruct a large size bone defect has not been developed yet. A thin Ti microfiber mesh scaffold is an osteoconductive material that can potentially be used as a reconstruction device for bone [1,2]. Furthermore, these materials are non-absorbable and can be processed into any desired shape. Thus, they are expected to be useful three-dimensional scaffolds for bone reconstruction. Compared with porous *β*-tricalcium phosphate scaffolds, porous Ti microfiber scaffolds demonstrate osteoblast differentiation, mineralization, and production of osteoprotegerin for osteoclast differentiation of human mesenchymal stromal cells [3]. Thus, Ti microfiber scaffolds enable rapid and stable bone formation and integration while maintaining the shape of the bone reconstruction site. We previously reported that thin Ti microfiber scaffolds restored bone continuity of a segmental and critical-sized mandibular defects in rabbits [4]. However, it was reported that the bone formation ratio inside of the device was less than 50%, indicating that osteoconductivity into the device should be improved. Ultraviolet (UV)-mediated photofunctionalization of Ti surfaces enhances osteoblast behavior and strengthens osseointegration [5]. UV-treated Ti surfaces are superhydrophilic because chemical deposition of hydrocarbon is cleaned [6,7]. Osteoblasts on UV-treated Ti surfaces extend their cytoskeletons with expression of cell adhesion markers [8]. UV-photofunctionalization is more effective on micro-roughened Ti surfaces [5], promoting osteoblast differentiation and rapid osseointegration [9,10,11]. Photofunctionalized dental implants improved a risk of early failure [12] and promised faster and stable osseointegration even in extremely poor conditions [13].

Here, we hypothesized that UV treatment and/or acid-etching treatment of Ti microfiber-based material improves their osteoconductivity. Improving hydrophilicity enables osteoblasts to infiltrate into the scaffold, and acid-etched Ti microfiber surfaces enhance osteoblast functional activity. The present study aimed to analyze the effect of acid-etching and UV treatment on osteoblast behavior in the Ti microfiber scaffold. Hydrophilicity of the scaffold and cell attachment, alkaline phosphatase (ALP) activity, and mineralization of osteoblasts derived from the femur and mandibular bone marrow were evaluated.

## 2. Results

### 2.1. Surface Morphology of Ti Microfiber Scaffold

Ti microfibers (125 µm diameter) were sintered into a disk-like scaffold structure 10 mm in diameter and 1 mm thick (Figure 1a–d). Scanning electron microscopy (SEM) images showed that the diameter of the original Ti fibers was 125 µm (Figure 1e,g) and acid-etched Ti microfibers was 100 µm (Figure 1f,h). The average weights of the original and acid-etched scaffolds were 0.11 and 0.08 g, respectively. The average porosity of the acid-etched Ti microfiber scaffold was approximately 80%, while that of the original Ti microfiber scaffold was approximately 70%. The average minimum and maximum pore sizes were approximately 20 and 500 µm, respectively, in both the original and acid-etched scaffolds. The surface morphology of the original Ti microfiber was relatively smooth (Figure 1i), whereas the acid-etched Ti microfiber had a micro-roughened surface (Figure 1j).

### 2.2. Newly Created Hydrophilicity of Ti Microfiber Disks

To identify the hydrophilicity/hydrophobicity of the Ti microfiber scaffold, a 5 µL droplet of water was placed on the scaffold. A droplet on the untreated scaffold of the original and acid-etched Ti microfibers remained in droplet form (Figure 2a,c), whereas the droplet spread into the UV-treated scaffold (Figure 2b,d). The contact angle of a droplet and the scaffold surface was approximately 70% in the untreated scaffold, whereas no contact angle was seen in both original and acid-etched scaffolds with UV treatment (Figure 2e). The water spreading area on the surfaces of both original and acid-etched scaffolds with UV treatment were significantly greater than those without UV treatment (Figure 2f).

Additionally, to identify liquid infiltration into the scaffolds, blue-colored hypertonic liquid made of diluted glycerol was injected into a well, in which the acid-etched disks were placed in the center (Figure 2g,h). The liquid did not infiltrate into the untreated disk, but rather lifted and moved the disk to the wall of the well (Figure 2g). In contrast, the liquid infiltrated into the UV-treated Ti fiber disk. The disk soaked the liquid and kept the position of center in the well while the liquid infiltrated (Figure 2h).

### 2.3. Bone Marrow-Derived Osteoblast Attachment and Functional Activity

Attachment of rat bone marrow-derived osteoblasts to the Ti microfiber scaffolds was examined after 3 and 24 h cell seeding. The number of osteoblasts attached to UV-treated original and acid-etched Ti microfibers was significantly greater than that attached to untreated Ti microfibers except at 3 h in the acid-etched Ti microfibers (Figure 3a). The number of cells attached on the UV-treated original Ti microfibers at 24 h was remarkable. Confocal laser microscopy showed fewer osteoblasts attached to the microfibers at 3 h (Figure 3b–e). At 24 h, osteoblasts on the UV-treated original Ti microfibers were prominent compared with those of the untreated Ti microfibers (Figure 3f,g), and osteoblasts on the UV-treated acid-etched Ti microfibers were prominent compared with those of the untreated Ti microfibers (Figure 3h,i). 

Confocal microscopy of osteoblasts with dual staining of actin and vinculin showed that the number of attached osteoblasts with actin and vinculin expression on the original and acid-etched microfibers with UV treatment were greater than those without UV treatment at 3 h (Figure 4). At 24 h, the number of attached osteoblasts increased and spread more on the original and acid-etched Ti microfibers with UV treatment (Figure 5). 

ALP activity and mineralization of the osteoblasts were prominent on the UV-treated Ti microfibers (Figure 6). ALP activity in the UV-treated Ti microfiber scaffolds was significantly greater than that in the untreated Ti fiber scaffold at Days 5 and 8 of culture in both the original and acid-etched Ti microfibers. Acid-etched Ti microfiber scaffolds showed higher ALP activity than that of the original Ti microfiber scaffolds (Figure 6a). Mineralization on UV-treated Ti microfiber scaffolds was significantly greater than that in the untreated Ti microfiber scaffolds at Days 8 and 15 of culture in both the original and acid-etched Ti microfibers (Figure 6b).

### 2.4. Mandibular Osteoblastic Function

Mandibular-derived osteoblasts seeded on the acid-etched Ti microfiber scaffolds were also examined in vitro. The number of osteoblasts attached to the UV-treated Ti microfiber scaffolds was significantly greater than that to the untreated Ti microfiber scaffolds at 3 and 24 h (Figure 7a). ALP activity of osteoblasts attached to the UV-treated Ti microfiber scaffolds was significantly greater than those attached to the untreated Ti microfiber scaffolds at Days 4 and 8 (Figure 7b). 

### 2.5. Surface Chemistry of Ti Microfibers

X-ray photoelectron spectroscopy (XPS) survey spectra showed a high intensity of atomic carbon on acid-etched Ti microfibers before UV treatment. The intensity of carbon on this surface was approximately 80%, while atomic oxygen and Ti were approximately 10% and 1%, respectively. The intensity of carbon was reduced to approximately 60%, and oxygen and Ti were increased to approximately 20% and 5%, respectively, after UV treatment (Figure 8a,b). 

## 3. Discussion

The present results reveal that UV treatment enhanced osteoblast behavior in Ti microfiber scaffolds and that acid etching treatment has advantages for ALP activity and mineralization, which were more prominent by further UV treatment. Superhydrophilicity as seen in UV-treated scaffolds seems to play an important role in enhancing osteoblast attachment and consequently its behavior. Ti surface time-dependently loses hydrophilicity after the surface preparation because of the progressive accumulation of oxygen-containing hydrocarbon [6,14]. This phenomenon is called “biological aging of titanium surface” [15] and the accumulation can be cleaned by UV treatment [6], even in Ti microfiber scaffolds, which are similar to the present scaffold [16]. The results of XPS analysis show that exposure to UV light reduced the amount of carbon on the untreated Ti microfiber surface. We previously reported that UV treatment removed hydrocarbons that had accumulated on the Ti surface, which dramatically reduced the amount of carbon on the surface [6,17]. UV light breaks the bonds between oxygen atoms in TiO_2_ and carbon atoms in hydrocarbons [6,18]. However, compositional analysis in this study was performed using XPS. XPS is more appropriate for the analysis of components on relatively large specimen surfaces than other techniques, such as Fourier-transform infrared spectroscopy (FTIR) and Raman spectroscopy. Thin Ti microfibers are small samples for the evaluation of surface composition; therefore, detailed analysis should be performed in a future study. The commercially available Ti microfibers used in this work comprised pure titanium. Therefore, the only elements that should have been present on their surfaces were Ti and O as titanium oxide, and accumulated carbon. We previously reported that Ti, O, and C accounted for greater than 99% of the surface elements on an acid-etched sample [18]. 

In the present study, a hypertonic liquid, which was assumed to have the same viscosity as blood, was used to evaluate the hydrophilicity of scaffolds placed in the middle of the plate. The untreated Ti microfiber scaffold was pushed by liquid pressure, whereas the liquid smoothly infiltrated into the UV-treated scaffold, indicating that the culture medium easily infiltrated into the scaffold and, consequently, osteoblasts attached to the Ti microfibers. As-made Ti fibers are too thin for osteoblasts to attach, spread, and function for mineralization. Additionally, as-made Ti microfibers are hydrophobic to hydro-repellent because of carbon contamination, which is even worse for cell attachment and spreading. UV treatment cleans this carbon contamination on the Ti microfiber surface. The decarbonized Ti microfiber becomes a superhydrophilic surface, which attracts and enables osteoblasts to attach and spread even on these thin and rounded fibers (Figure 9). Osteoblast activity on superhydrophilic Ti surfaces by UV treatment is greater than that on saline-stored Ti surfaces [19]. Osteoblast attachment on the Ti surface is established within at most 24 h [20,21]. The attached osteoblasts spread to start proliferating and functioning with cellular cytoskeletal formation as well as the expression of the focal adhesion protein vinculin. Vinculin promotes extracellular matrix–cytoskeletal interactions and regulates cell survival [22,23]. The present results reveal that vinculin expression was remarkable in the scaffold with UV treatment, suggesting that UV treatment enhanced osteoblast attachment on the Ti microfibers. In particular, the UV-treated scaffolds made of acid-etched Ti microfibers showed prominent expression of vinculin at 3 h. It was thought that this rapid establishment of osteoblast attachment compared to other scaffolds resulted in higher ALP activity and mineralization. We previously reported that osteoblasts on UV-treated thin Ti fibers spread widely with enhanced expression of vinculin [16] and that vinculin expression inhibited early apoptosis of the osteoblasts [18]. UV treatment converts the electrical charge of Ti surface to a positive charge, which allows a direct connection between the cells and Ti [8]. Electrostatic conditions influence the connection between cells and material surfaces. A positively charged surface is preferable for cell attachment because cells are essentially negatively charged. An untreated Ti surface is negatively charged; therefore, it requires modification before cells can bind to it. However, cells can directly attach to positively charged surfaces [8]. This advantageous effect of UV-photofunctionalization of the Ti surface possibly facilitates osteoblast attachment on the Ti surface, even under poor conditions of thin and rounded Ti microfiber scaffolds.

Surface modification influences osteoblastic behaviors, such as proliferation and mineralization. Hirota et al. [24] reported that osteoblasts attached on hydroxyapatite (HA)-coated thin Ti fibers accelerated their functional activity for mineralization. In contrast, osteoblasts attached to untreated thin Ti fibers proliferated. In the present study, ALP activity and mineralization were more prominent on the acid-etched scaffolds, although the number of attached cells was greater on the original scaffolds. These results appear contradictory. The results of our previous study also indicated that osteoblast proliferation was enhanced on untreated Ti surfaces rather than UV-treated surfaces. The ALP activity and mineralization of osteoblasts attached to UV-treated Ti surfaces were enhanced relative to cells on untreated Ti surfaces [5]. Osteoblasts generally start functioning once they stop proliferating. In contrast, our results indicate that osteoblasts on UV-treated Ti scaffold surfaces tended to initiate function and differentiating processes rather than proliferation. We previously reported that the differentiation of mesenchymal stem cells into osteoblasts was more rapid on UV-treated surfaces than on untreated surfaces [25]. However, ascorbic acid and *β*-glycerophosphate were used in the previous work and in this study, which induced osteoblast activity and differentiation. This indicated that the behavior of osteoblasts observed in the present study may have been due to those factors. Additionally, ALP activity on the acid-etched surfaces was highly dispersed on Day 8 of this study. Although the value may not have been precise, this suggested that there was a statistically significant difference between the acid-etched surfaces and the control. Because these were the results of a pilot study, further investigation is needed to confirm the enhancement of osteoblast activity on acid-etched surfaces at a minimum of three time points. These should include the initial stage for ALP activity and the late stage for mineralization. 

Thin Ti microfibers are considered to be a difficult environment for the attachment and growth of osteoblasts because the surface is narrow and round. In other words, osteoblasts cannot effectively attach to thin microfibers. Therefore, the number of osteoblasts attached to microfibers should be increased to enhance bone formation in the scaffold. The superhydrophilic state of the scaffold obtained by UV treatment is more likely to attract osteoblasts on Ti microfibers. UV treatment also enables osteoblasts to spread everywhere in the scaffold, in which the development of cellular cytoplasmic projections and cytoskeletal formation, as well as the expression of focal adhesion proteins, were also enhanced. Only the hydrophilicity of the scaffold surfaces was evaluated in this study. The water-droplet or Byreck method should be used to evaluate water absorbency in future studies.

Surface topography also influences osteoblastic behavior. The effect of acid etching was more prominent on ALP activity and mineralization. Surface topography produced with acid etching accelerates osteoblast activity and mineralization in vitro [26]. In addition, the hardness and stiffness of bone tissue produced on roughened Ti surfaces in vivo are equivalent to those of untreated cortical bone [27]. The present results demonstrate that these effects of acid-etching treatment are effective even on thin Ti microfibers. Remarkably greater ALP activity was seen in mandibular-derived osteoblast as well, indicating this enhanced affinity of osteoblasts on the Ti microfiber surface by UV treatment was versatile.

Use of UV-photofunctionalization on acid-etched thin Ti microfibers is expected to enhance bone formation deeply into Ti microfiber scaffolds. Previous studies reported that UV-photofunctionalization enhanced not only osteoblastic behavior and mineralization but also in vivo bone formation on the Ti mesh [28,29]. Bone engineering into the Ti microfiber scaffolds, which have more complicated three-dimensional structure than Ti mesh, are still a challenging subject. Further in vivo investigations of actual bone formation into thin Ti microfiber scaffolds with mechanical and histological evaluation are needed to explore optimal conditions of the material. Achieving secure and stable osteoblast affinity and function on thin Ti microfibers tuned with UV photofunctionalization could lead to developing bone engineering using Ti microfiber scaffolds. 

## 4. Materials and Methods 

### 4.1. Titanium Microfiber Scaffold Characterization and UV-Photofunctionalization

Thin and round commercially available pure Ti fibers (99.6%, Sigma-Aldrich, St. Louis, MO, USA), 125 µm in diameter, 2000 mm long were woven together to form disks approximately 10 mm in diameter and 1 mm thick.

To roughen some of the samples, scaffolds were introduced by acid-etching with 67% (*w*/*w*) H_2_SO_4_ (Sigma) at 120 °C for 10 s. The diameters and thicknesses of the original and acid-etched samples were calculated using ImageJ program (NIH, Bethesda, ML, USA). The diameters and thicknesses were then used to calculate the total volumes of the scaffolds. The porosity of untreated scaffolds was calculated by subtracting the Ti microfiber volume from the total volume. The porosity of the acid-etched scaffolds was calculated by comparing their weight ratio to those of the untreated scaffolds. The samples were autoclaved, and then placed and stored at a dark condition for four weeks to obtain biological aging. The surface morphologies of the Ti microfibers were examined by SEM (Nova 230 Nano SEM, FEI, Hillsboro, OR, USA). The minimum and maximum pore sizes of the Ti microfibers were determined by measuring 30 pores in the image. Photofunctionalization was performed by treating Ti microfiber scaffolds with UV light for 15 min using a photo device (TheraBeam Affiny; Ushio Inc, Tokyo, Japan) immediately before use.

### 4.2. Hydrophilicity of Ti Microfiber Scaffolds

The hydrophilicity of the scaffold was examined by contact angle and spreading area of a-5 µL diluted distilled H_2_O (ddH_2_O) droplet. The contact angle between the droplet and the scaffold surface was measured by a contact angle meter (CA-X, Kyowa Interface Science, Tokyo, Japan). The spreading area on the surface was defined as [(H_2_O area/total scaffold surface area) × 100] (%) and measured using ImageJ (NIH). The mean values of both contact angle and spreading area on the original and acid-etched scaffolds with or without UV treatment were calculated. A hypertonic glycerol solution (Wako, Tokyo, Japan) was diluted three times. The diluted glycerol solution was assumed to have the same viscosity as that of blood [30]. The solution was mixed with an aqueous methylene blue solution (Wako) for visualization to determine whether the liquid could penetrate into the scaffold. The scaffold was placed in the center of a well of a 12-well plate, and 1 mL of the liquid was poured into the well. The motion of the scaffold against the hypertonic infiltration was observed.

### 4.3. Osteoblastic Cell Culture

Bone marrow-derived osteoblastic cells were isolated from the femurs and mandibles of eight-week-old male Sprague–Dawley rats and placed into alpha-modified Eagle’s medium supplemented with 15% fetal bovine serum, 50 mg/mL ascorbic acid, 10 mM Na-*β*-glycerophosphate, 10^−8^ M dexamethasone, and antibiotic–antimycotic solution containing 10,000 units/mL penicillin G sodium, 10,000 mg/mL streptomycin sulfate, and 25 mg/mL amphotericin B. The cells were incubated in a humidified atmosphere of 95% air and 5% CO_2_ at 37 °C. At 80% confluency, cells were detached using 0.25% trypsin-1 mM EDTA-4Na and seeded onto Ti microfiber scaffolds placed in a 24-well culture dish at a density of 2 × 10^4^ cells/cm^2^. The scaffolds were affixed onto the plate using Super-Bond C&B adhesive acrylic resin (SUN MEDICAL, Shiga, Japan). The culture medium was renewed every three days.

### 4.4. Cell Attachment Assay and Osteoblast Behavior on Ti Microfiber

Cell attachment of osteoblasts was evaluated by measuring the number of cells attached to the Ti microfiber scaffolds after 3 and 24 h of seeding. The numbers of viable cells were quantified using a WST-1-based colorimetric assay (WST-1, Roche Applied Science, Mannheim, Germany). A culture well was incubated at 37 °C for 1 h with 100 μL of WST-1 reagent. The amount of formazan product was measured using a multi-detection microplate reader (Synergy^TM^ HT, BioTek Instruments, Inc., Winooski, VT, USA) at a wavelength of 450 nm. Osteoblasts on Ti microfibers were also morphometrically observed using confocal laser scanning microscopy. Cells were fixed in 10% formalin after culture for 3 and 24 h. The cells were stained with the fluorescent dye rhodamine phalloidin (actin filament, red color, Molecular Probes, Eugene, OR, USA). Expression of the cell adhesive protein vinculin was examined with double staining using a mouse anti-vinculin monoclonal antibody (Abcam, Cambridge, MA, USA), followed by FITC-labelled anti-mouse secondary antibody (Abcam). 

### 4.5. ALP Activity and Mineralization Assay

ALP activity and mineralization activity on Days 5 and 8 were examined. For ALP activity, the cultures were rinsed with ddH_2_O, and then 250 mL of *p*-nitrophenylphosphate was added. The samples were then incubated at 37 °C for 15 min. The ALP activity was evaluated as the quantity of nitrophenol, and measured at a wavelength of 405 nm using a plate reader. The mineralization capabilities of osteoblasts were evaluated by quantifying a calcium deposition. The cultures were washed with phosphate buffered saline and incubated in 1 mL of 0.5 mol/L HCL solution overnight with gentle shaking. The samples were mixed with *o*-cresolphthalein complexone in an alkaline medium (Calcium Binding and buffer Reagent, Sigma) to produce a red calcium–cresolphtalein complexone complex. The color intensity was measured using a plate reader at 575 nm.

### 4.6. XPS Analysis

XPS (PHI Quantera SXM, UPVAC-PHI, Inc, Kanagawa, Japan) was performed to determine the atomic distribution of the Ti microfibers. Carbon, Oxygen, and Titanium amounts on the scaffold with or without UV treatment were evaluated.

### 4.7. Statistical Analysis

Hydrophilic evaluation and culture studies were performed in triplicate (*n* = 3). The *t*-test was used to evaluate differences between untreated and UV-treated groups after confirming normal distribution in each group. *p*-values of < 0.05 were considered significant.

## 5. Conclusions

UV treatment converts the Ti microfiber surface to superhydrophilic and significantly removes hydrocarbons, which allows osteoblasts to attach to the surface and spread, leading to the manifestation of normal osteogenic function even on this thin and rounded microfiber. The number of cells attached to the Ti microfibers was 3–10 times greater when the scaffolds were treated with UV. The development of cytoplasmic projections and cytoskeletal as well as the expression of focal adhesion protein were exclusively observed on UV-treated scaffolds However, a more detailed analysis of cell diameters and areas as well as the quantification of vinculin expression should be performed in future studies. Osteoblastic function was 2–15 times greater on UV-treated scaffolds, with more pronounced enhancement on the acid-etched Ti microfiber scaffolds. Tuning of Ti microfiber scaffolds with UV-photofunctionalization enhances osteoblastic behavior, which is proposed as a new strategy for bone engineering.

## Figures and Tables

**Figure 1 ijms-21-00738-f001:**
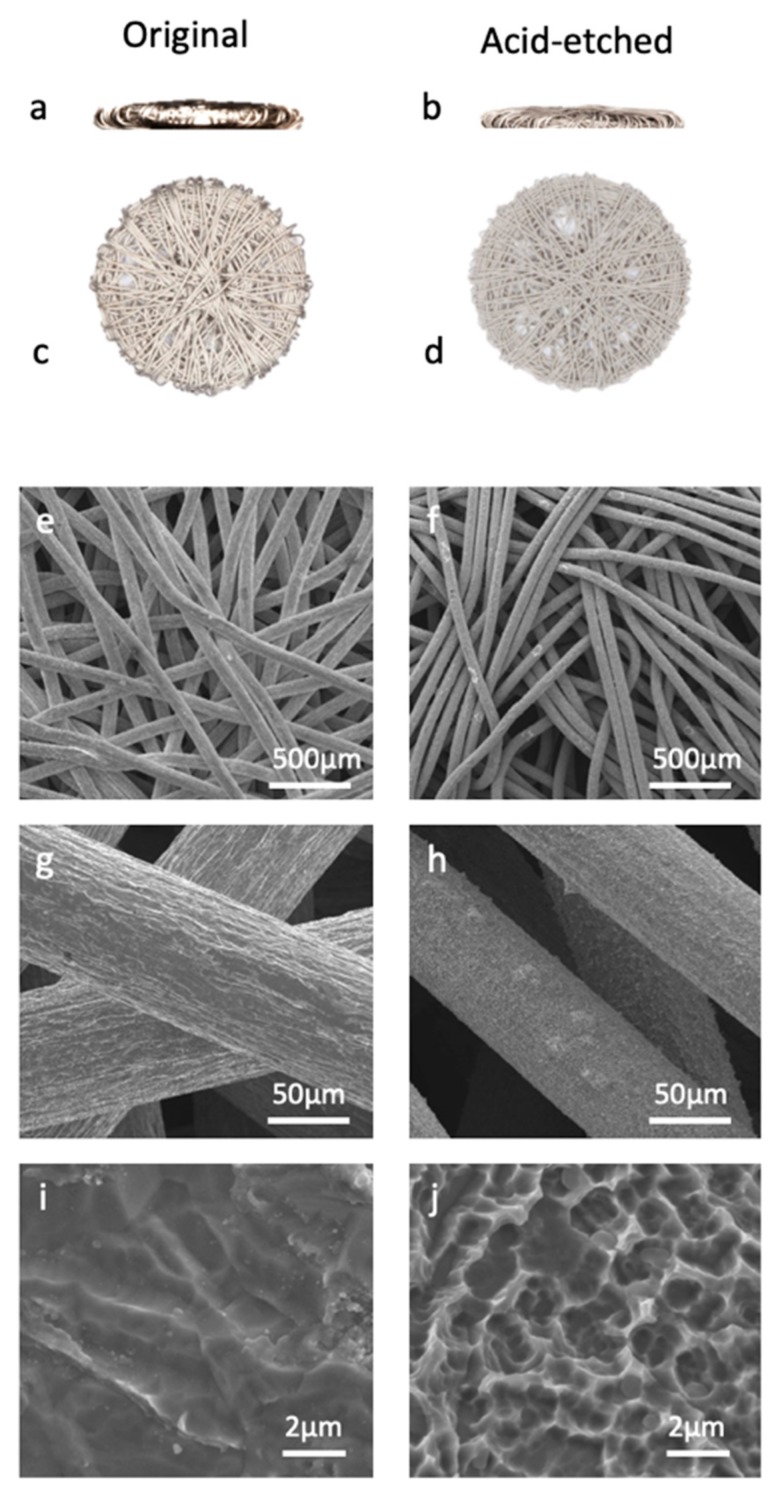
Creation of Ti microfiber scaffolds and morphology of the surface. Top view of (**a**) original and (**b**) acid-etched disks, and side view of (**c**) original and (**d**) acid-etched disks. Scanning electron microscopy images of the original Ti microfibers (**e**,**g**,**i**) and the acid-etched Ti microfibers (**f**,**h**,**j**).

**Figure 2 ijms-21-00738-f002:**
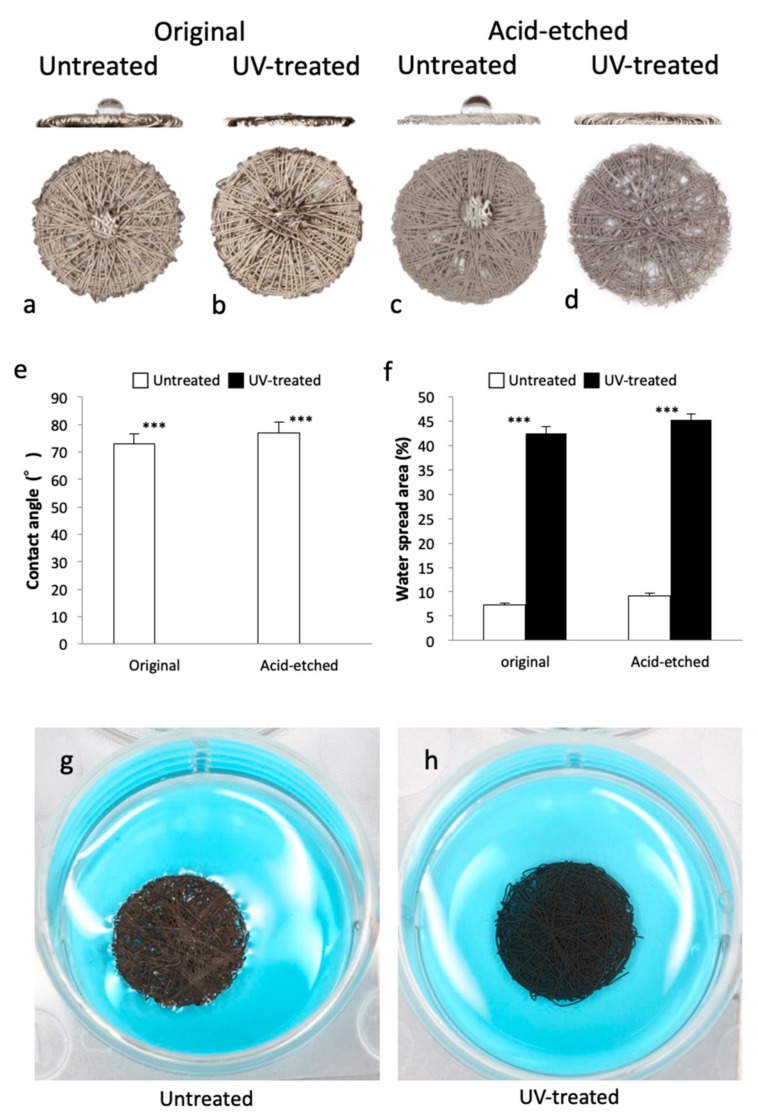
Hydrophilicity/hydrophobicity of the scaffolds (**a**–**d**), contact angle (**e**) and spread areas (**f**) of water droplets on the scaffolds. Lateral views indicate contact angle between surface and water droplets. Water droplets remain on the surface of the untreated scaffolds in (**a**,**c**), whereas water droplets are completely absorbed by the UV-treated scaffolds in (**b**,**d**). Lateral views of the UV-treated scaffolds show no contact angles between the water droplets and the surfaces. Infiltrability of the untreated (**g**) and UV-treated (**h**) original Ti microfiber scaffolds. The untreated disk does not soak in the liquid, whereas the UV-treated disk does soak up the liquid. The untreated disk is pushed to the wall of the well by the liquid. Each value represents the mean ± standard deviation of triplicate experiments (*n* = 3). *** *p* < 0.001

**Figure 3 ijms-21-00738-f003:**
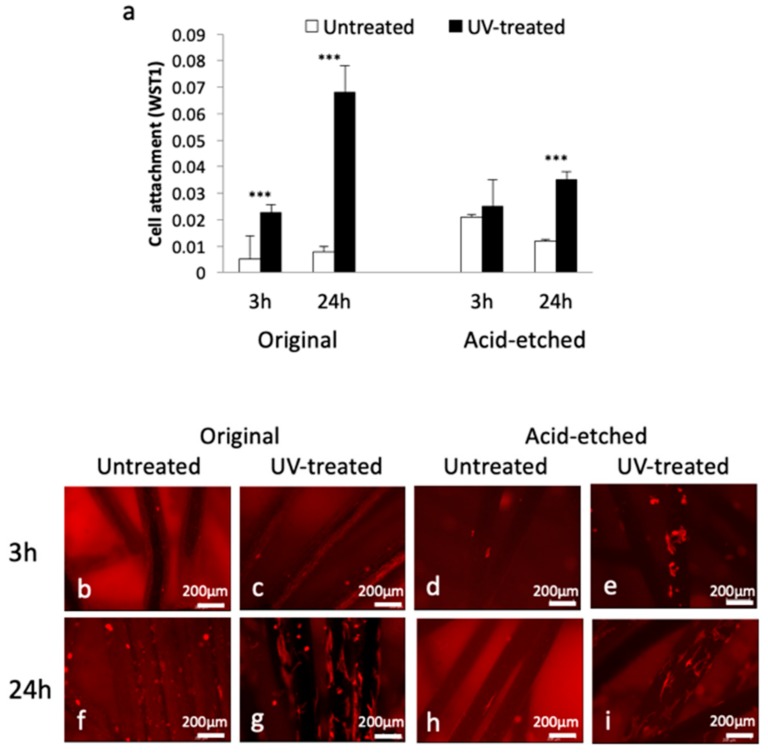
Cell attachment of rat bone marrow-derived osteoblasts on original/acid-etched, and untreated-/UV-treated Ti microfiber scaffolds. (**a**) The number of cells attached to each scaffold with or without UV treatment at 3 and 24 h, as evaluated by WST-1 assay. The number of cells in the UV-treated Ti microfiber scaffolds was significantly greater than that in the untreated Ti fiber scaffolds except for acid-etched scaffolds at 3 h. (**b**–**i**) Microscopic appearance. Small, round shaped cells sparsely attach on the untreated Ti microfibers at 3 h (**b**–**d**), and on those at 24 h (**f**,**h**). Relatively many cells attach on the UV-treated acid-etched Ti microfibers (**e**). The shape of osteoblast spreads and the number of cells increases in the UV-treated Ti microfibers at 24 h (**g**,**i**). Each value represents the mean ± standard deviation of triplicate experiments (*n* = 3). *** *p* < 0.001.

**Figure 4 ijms-21-00738-f004:**
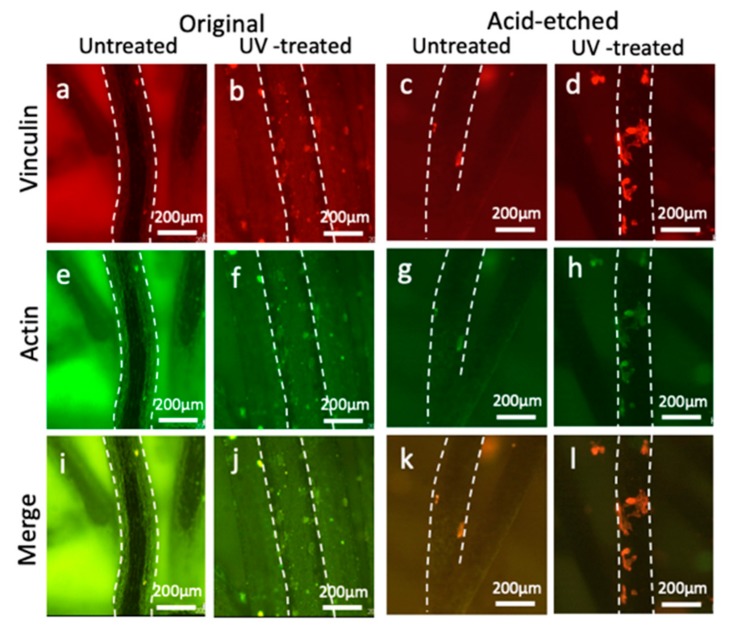
Vinculin and actin staining of osteoblasts on original/acid-etched and untreated/UV-treated Ti microfibers at 3 h: vinculin staining (**a**–**d**); actin staining (**e**–**h**); and merged images (**i**–**l**). Small, round shaped cells sparsely attach on each Ti microfiber. The number of attached osteoblasts on UV-treated microfibers was greater than that on untreated microfibers on both original and acid-etched Ti microfibers. The shape of the osteoblasts on UV-treated acid-etched Ti microfibers (**d**,**h**,**l**) spreads and is larger than that on UV-treated original Ti microfibers (**b**,**f**,**j**). Dotted white lines show the edges of the fibers.

**Figure 5 ijms-21-00738-f005:**
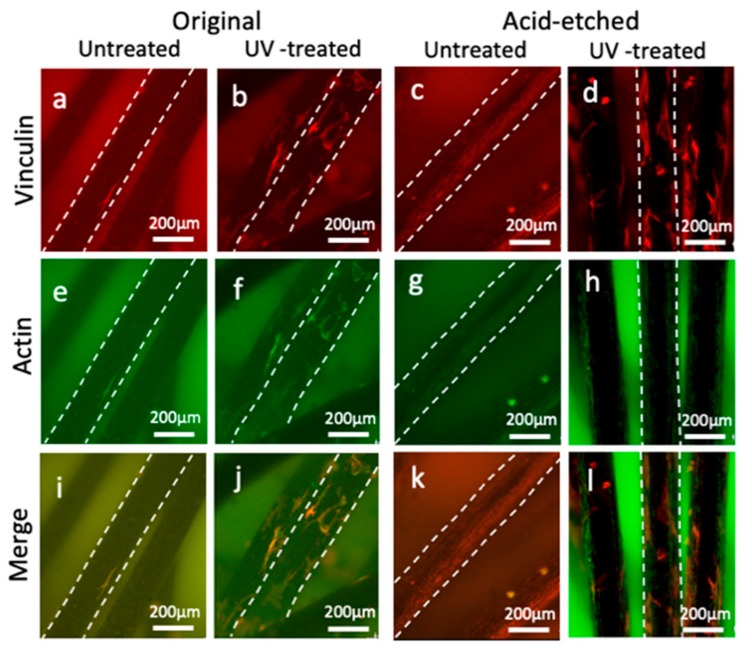
Vinculin and actin staining of osteoblasts on original/acid-etched and untreated-/UV-treated Ti microfibers at 24 h: vinculin staining (**a**–**d**); actin staining (**e**–**h**); and merged images (**i**–**l**). Small, round shaped cells sparsely attach on each Ti microfiber. The shape of the osteoblasts on UV-treated original (**b**,**f**,**j**) and acid-etched (**d**,**h**,**l**) Ti microfibers spreads with both vinculin and actin expressions on the fibers. Dotted white lines show the edges of the fibers.

**Figure 6 ijms-21-00738-f006:**
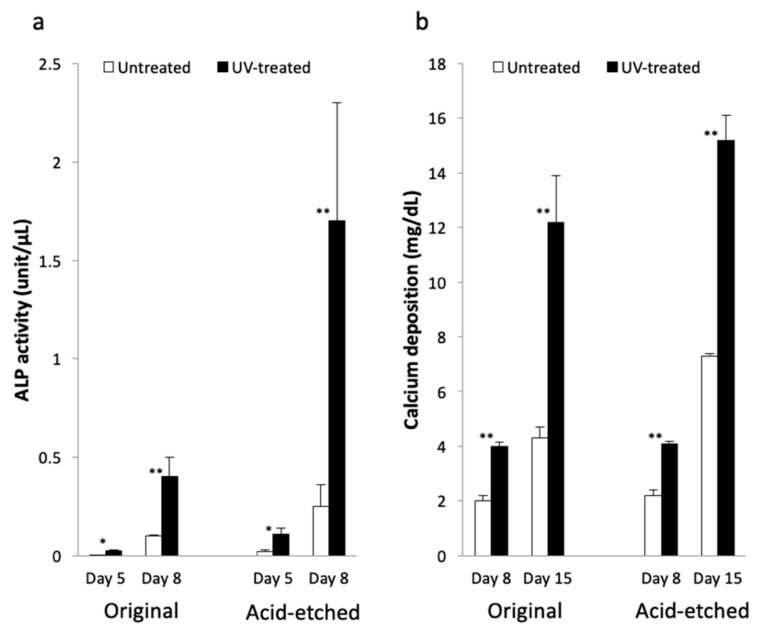
Results of ALP activity and mineralization. (**a**) ALP activity of the original and acid-etched Ti microfiber scaffolds with or without UV-treatment at Days 5 and 8 of culture. ALP activity of osteoblasts in the UV-treated Ti microfiber scaffolds is significantly greater than that of the untreated Ti microfiber scaffolds. (**b**) Mineralization of original and acid-etched Ti microfiber scaffolds with or without UV-treatment at Days 8 and 15 of culture. Mineralization of osteoblasts in the UV-treated Ti microfiber scaffolds is significantly greater than that of untreated Ti microfiber scaffolds. Each value represents the mean ± standard deviation of triplicate experiments (*n* = 3). * *p* < 0.05, ** *p* < 0.01.

**Figure 7 ijms-21-00738-f007:**
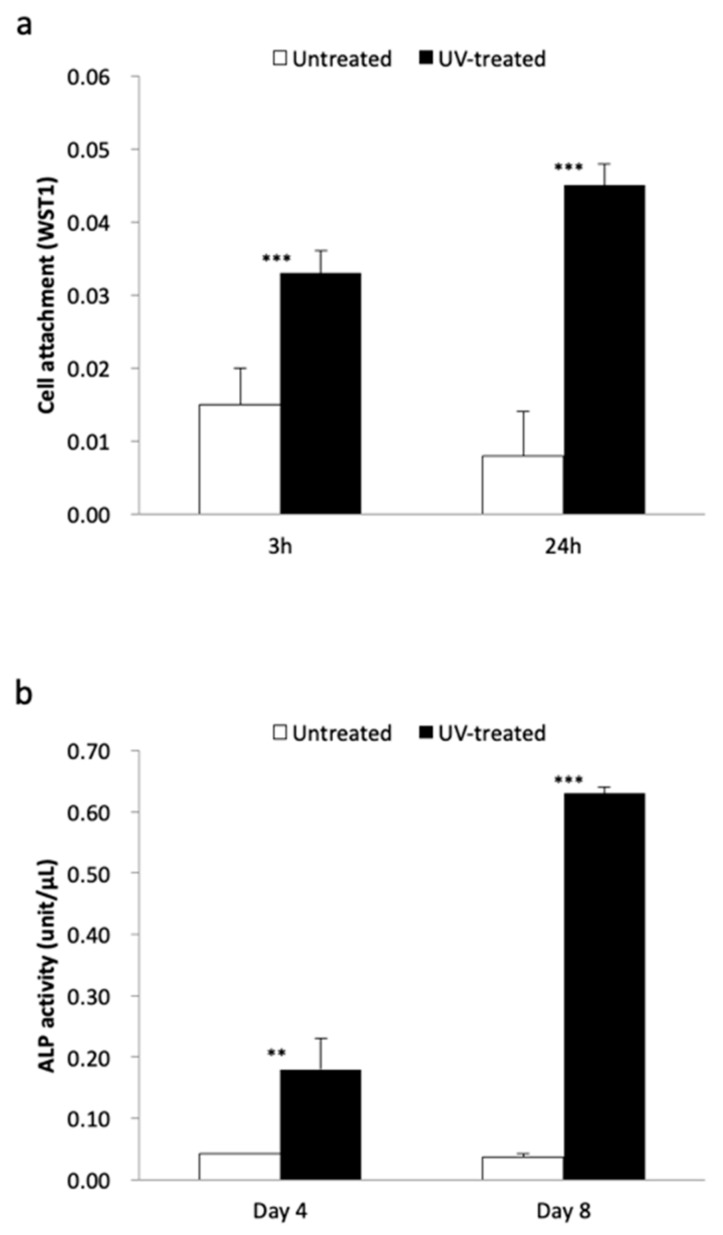
(**a**) Number of osteoblasts attached to the UV-treated Ti fiber scaffold disks is significantly greater than that to the untreated Ti fiber scaffold disks at 3 and 24 h. (**b**) ALP activity of osteoblasts attached to the UV-treated Ti fiber scaffolds is significantly greater than those attached to the untreated Ti fiber scaffolds at Days 4 and 8. Each value represents the mean ± standard deviation of triplicate experiments (*n* = 3). ** *p* < 0.01, *** *p* < 0.001.

**Figure 8 ijms-21-00738-f008:**
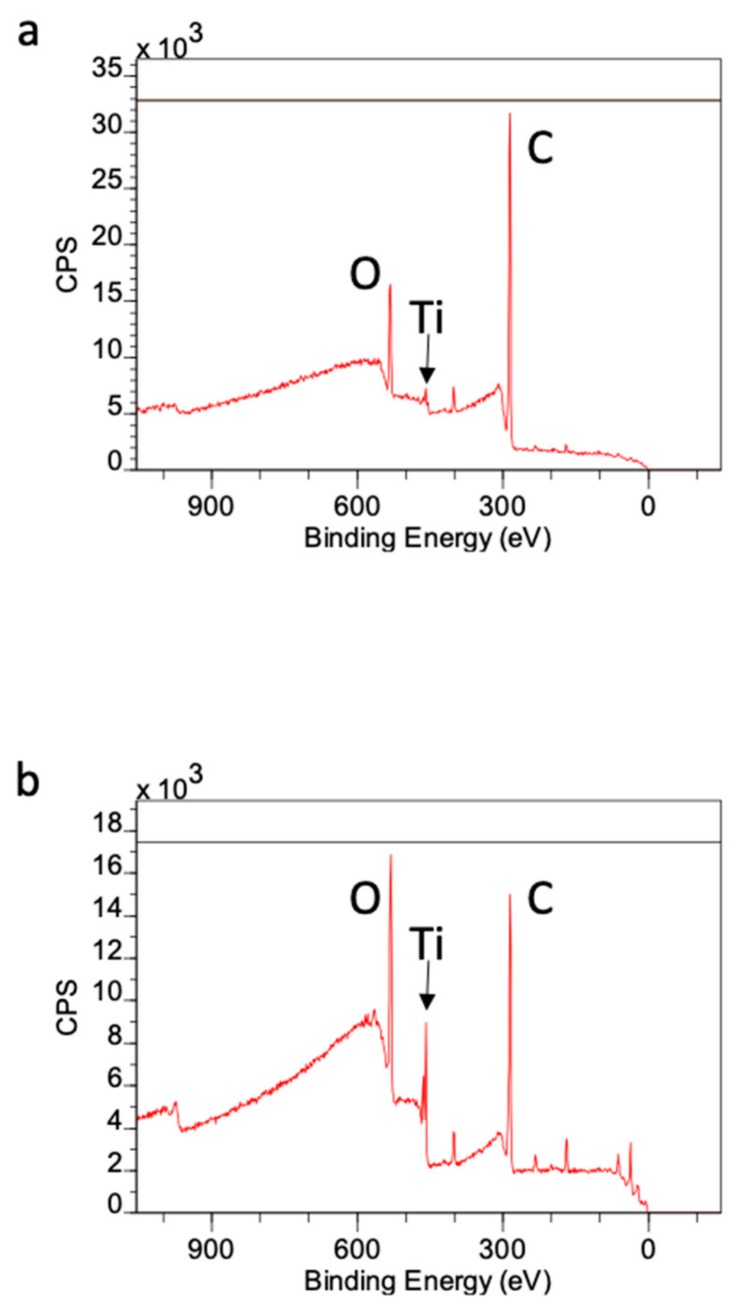
(**a**) XPS survey spectra of acid-etched Ti microfiber scaffold. The intensity of carbon is prominent. (**b**) XPS surgery spectra of the UV-treated acid-etched Ti microfiber scaffold. The intensities of oxygen and Ti is increased. O, oxygen; Ti, titanium; C, carbon.

**Figure 9 ijms-21-00738-f009:**
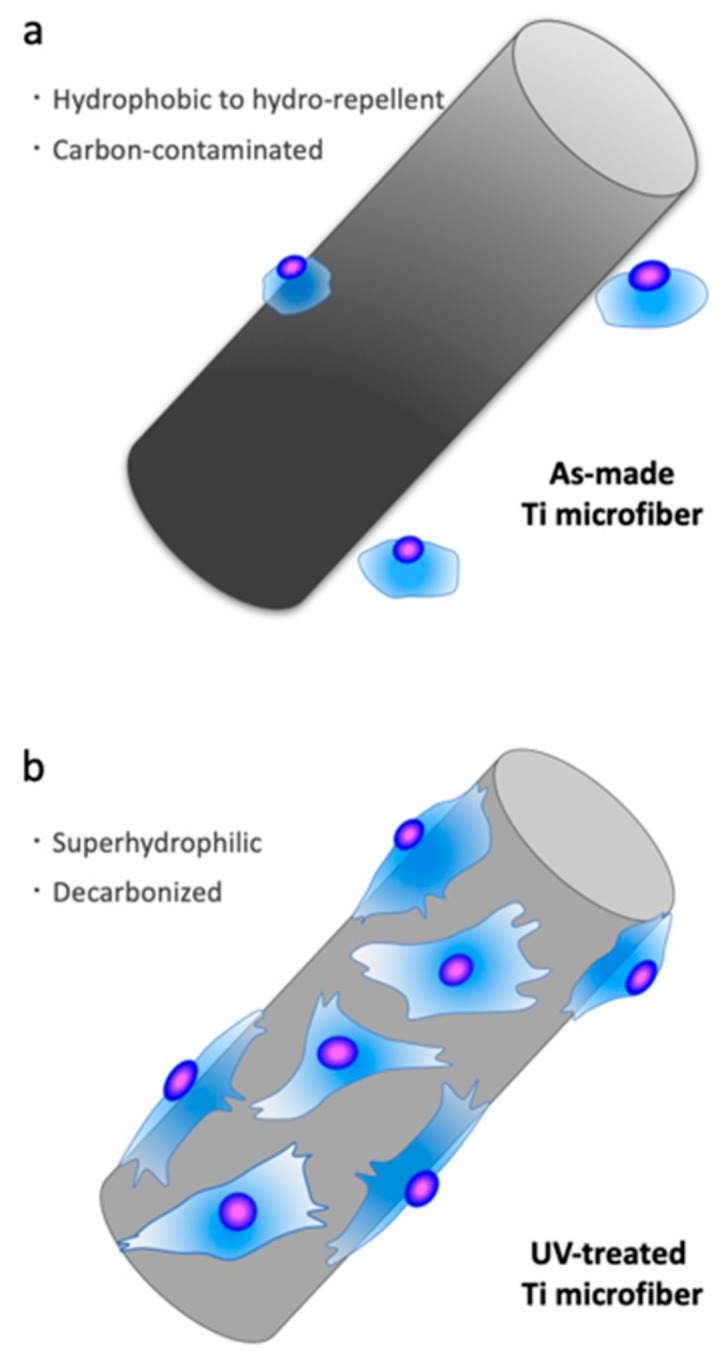
Schematic description of osteoblast behavior on Ti microfibers. (**a**) Ti microfibers are thin, rounded, hydrophobic/hydrorepellent, and inadvertently covered by hydrocarbon molecules, on which osteoblasts show extreme difficulty in attaching and setting. (**b**). In contrast, UV treatment converts the Ti microfiber surface to superhydrophilic and significantly removes hydrocarbons, which allows the osteoblasts to attach to the surface and spread, leading to the manifestation of normal osteogenic function even on this thin and rounded surface.

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
