# Peer review of "Tuning of Titanium Microfiber Scaffold with UV-Photofunctionalization for Enhanced Osteoblast Affinity and Function"

_ijms, 2020, doi:10.3390/ijms21030738_

Round 1
Reviewer 1 Report
The authors report about the effect of UV functionalization on osteoblast attachment on titanium microfibers. The manuscript is well written and has some structure to it. However, there are some information that requires further clarification.
Although titanium has been used alone and its alloys are also commonly investigated if the application is for dental or simple bone tissue engineering than these microfibres are non-degradable agents and alternatively would require a second stage surgery to remove them or they can be left in the defect site. Therefore, solid implants are more preferred than meshes. Then is the question of long term stability in-vivo. So it is still unclear why the microfibres were used.
Line 77; The porosity of the templates has been coated to be at 70% for neat and 80% for acid etched. How was this figure calculated ?. Image J, Fiji , how many pores were counted. alternatively, microCT can also reveal the porosity as well. Line 91 Hypertonic liquid was used to demonstrate and mimic body fluids. An ideal solution to mimic this would be blood itself, simulated body fluid. How much water was taken up by the mesh discs ?, Is there a way to quantify the super hydrophilicity of these de-carbonated specimens. Line 102 to 111 Figure 3 of cell viability shows that the original specimens that were UV treated had higher viability and the acid-etched and UV treated did not show higher attachment. Why is this contrast in the viability and Mineralized matrix deposition (ALP and Ca depo- day 8 and 15)? What does the ALP measurements look like on day 15? Is there more data after the 24 hours, like day 1, 3 and & 7-day rate. The mineralization was studied with these rat bone marrow osteoblasts after day 5 and 8. This is a rapid deposition of matrix rate the only reason to this could be that the media was supplemented with ascorbic acid and BGP, some osteogenic mediums were aided for this. The study requires further analysis of mineralized matrix deposition at least 3-time points without BGP and ascorbic acid. Line 151; Figure 6, Why is the acid etched samples UV treated on day 8 have an error margin that high. ? Line 170; XPS was done to analyze the carbon-free samples. There should be a more detailed analysis using either FTIR or Raman Spectroscopy with this. Line 215 and 216 "...treatment converts the electrostatic of Ti ......" the sentence needs rephrasing. Converts the charge? Line 224 ".....severe environment ....." what is a severe environment. ?? severe can be in a good sense as well this term needs clarification Line 247 "... on thin Ti micro......." perhaps a typo. Line 269; The seeding density was low considering the meshes were 70 to 80% porous. Usually, porous templates will allow cells to flush through them when the cell droplet is poured on top. How did the authors managed to do the seeding without a ring or was any other aid used to conduct cell culture? Line 305; 4.7. "Statistical analysisi" . another typo. What software was used to do the statistical analysis? Line309; The conclusion reports about the spread and cytoplasmic projections and cytoskeletal profiles of the attached cells. However, the confocal images are not overly clear to this claim, either the images should be redone with focus on cytoplasmic extensions and spreading profile of cells. Although UV functionalization has been reported previously the conclusion is not very clear and claims quoated are not supported by the data shown
Reviewer 2 Report
Reviewer:
<Journal Name> International Journal of Molecular Sciences
Manuscript ID: ijms-694024
Type of manuscript: Article
Title: Tuning of titanium microfiber scaffold with UV-photofunctionalization for enhanced osteoblast affinity and function
Authors: Chika Iwasaki, Makoto Hirota *, Miyuki Tanaka, Hiroaki Kitajima, Masako Tabuchi, Manabu Ishijima, Wonhee Park, Yoshihiko Sugita, Ken Miyazawa, Shigemi Goto, Takayuki Ikeda, Takahiro Ogawa
Reviewer's comment:
Dear Editor:
Reviewer's comment:
Dear Editor:
The manuscript focused on the “Tuning of titanium microfiber scaffold with UV-photofunctionalization for enhanced osteoblast affinity and function”. . It is very interesting and novel study in some fields. Based on above reason, it is recommended to publish it after major revision. However, some points need to revise, which is listed below
[1] The new relate references are needed to add in the revised manuscript.
[2] The authors investigate many parameters in this study. What is optimal condition in this work?
[3] What are the important applications in this study? Please add in the revised manuscript.
[4] The author should compare the specific properties of other materials and explain the benefits of this material.
[5] The porosity of the acid-etched Ti microfiber scaffold was approximately 80%. How to determine the porosity?
[6] The grammar and usage of English are needed to improve by Native English speaker.
[7] The authors should provide corresponding contact angle images in Fig. 2e.
[8] The compositional analysis should be done using adequate technique under untreated and UV-treated conditions.
[9] Mineralization of original and acid-etched Ti microfiber scaffolds with or without UV-treatment at day 8 and 15 of culture. How is determined to day 8 and 15 of culture?
[10] The intensity of carbon on this surface was approximately 80%, while atomic oxygen and Ti were approximately 10% and 1%, respectively. The intensity of carbon was reduced to approximately 60%, and oxygen and Ti were increased to approximately 20% and 5%, respectively, after UV treatment. Please explain why does this composition change.
Sincerely yours.
Round 2
Reviewer 1 Report
Dear Authors,
Thank you for the response. The manuscript now looks in a better format than before.
Regards
Reviewer.
Reviewer 2 Report
Dear Editor:
According to the revised version, it can be accepted and published in IJMS journal.
Sincerely yours.